# KAN OR MLP? POINT CLOUD SHOWS THE WAY FORWARD

## ABSTRACT

Multi-Layer Perceptron (MLP) has become one of the fundamental architectural component in point cloud analysis due to its simple, flexible structure and universality. However, when processing complex geometric structures in point cloud, MLP's fixed activation functions struggle to efficiently capture local geometric features, while suffering from poor parameter and computational efficiency. The high redundancy in deep MLP structures imposes constraints on the model's generalizability. In this paper, we propose PointKAN, which applies Kolmogorov-Arnold Network (KAN) to point cloud analysis tasks to investigate their efficacy in hierarchical feature representation. PointKAN adopts a hierarchical structure that progressively expands the receptive field, with each layer comprising three components: Geometric Affine Module (GAM), Local Feature Processing (LFP), and Global Feature Processing (GFP). The LFP stage employs a KAN structure to enhance feature extraction capabilities. Experimental results demonstrate that PointKAN outperforms PointMLP on benchmark datasets such as ModelNet40, ScanObjectNN, and ShapeNetPart. Notably, when handling large-scale point cloud data, PointKAN achieves significantly reduced computational overhead compared to PointMLP. Furthermore, its outstanding performance in few-shot learning tasks indicates that PointKAN exhibits stronger generalization capabilities than PointMLP. To further enhance parameter and computational efficiency, we develop Efficient-KAN in the PointKAN-elite variant. This work highlights the unique advantages of KAN over MLP in point cloud analysis and opens new avenues for research in point cloud understanding.

## 1 INTRODUCTION

Point cloud analysis is vital in computer vision and 3D processing, with applications spanning autonomous driving (Li et al., 2023) and robotics (Yan et al., 2020; Rusu & Cousins, 2011). Unlike structured 2D images, point clouds pose challenges due to their irregularity and sparsity. Recent deep learning advances (Zhang et al., 2022; Pang et al., 2022; He et al., 2024; Liang et al., 2024; Han et al., 2024) build upon pioneering MLP-based architectures like PointNet/PointNet++ (Qi et al., 2017a;b). Subsequent research, including PointNeXt (Qian et al., 2022), DGCNN (Wang et al., 2019), and Point Transformer (Zhao et al., 2021), employs CNN, GCN, and attention to capture local geometric features.

Notably, PointMLP (Ma et al., 2022) utilizes a pure MLP architecture for point cloud analysis. Its core design philosophy eschews complex convolutions or graph operations, instead relying on stacked residual MLP blocks for feature learning. While achieving competitive performance in classification and segmentation, PointMLP inherits inherent MLP limitations. Rooted in the Universal Approximation Theorem (Cybenko, 1989), MLPs employ linear weights and fixed activations. Consequently, capturing the geometric relationships in large-scale point cloud data requires an increase in the number of hidden layers, which inevitably leads to reduced computational efficiency and model redundancy.

In contrast, the recently proposed Kolmogorov-Arnold Network (KAN) (Liu et al., 2024) offers a promising MLP alternative, gaining significant attention for its accuracy and interpretability. KAN leverages the Kolmogorov-Arnold Representation Theorem (KART) (Arnold, 2009), which states that any multivariate continuous function can be decomposed into univariate functions. KAN re-

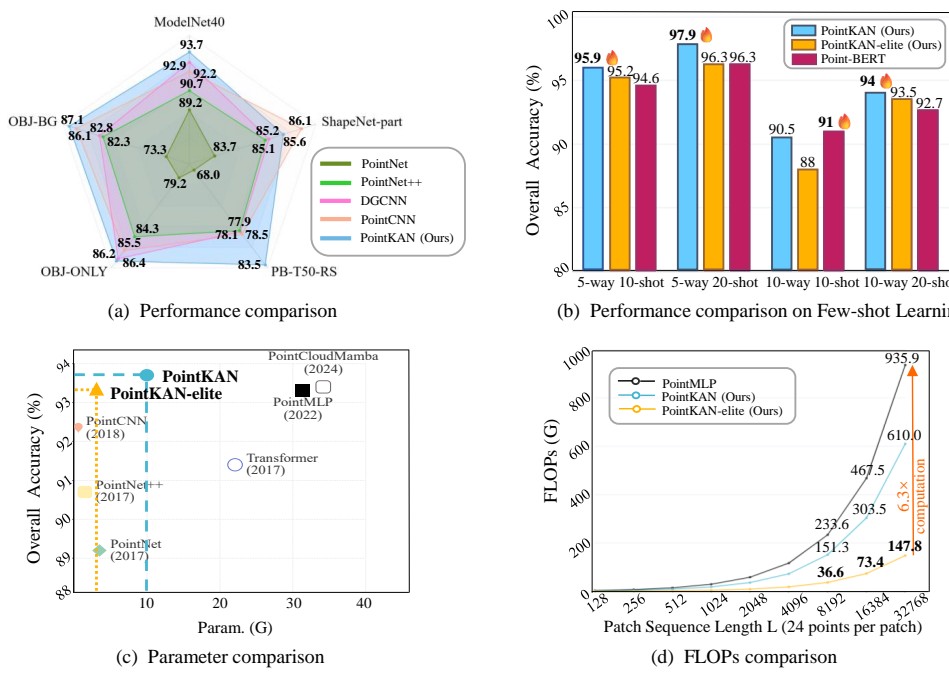

Figure 1: PointKAN (including PointKAN-elite) comprehensively outperforms counterparts. (a) Surpasses MLP/CNN baselines across datasets; (b) In Few-shot Learning, exceeds Point-BERT's pre-trained performance when trained from scratch; (c)-(d) Substantially reduces parameters versus PointMLP, with PointKAN-elite significantly curbing FLOPs growth as Patch Sequence Length increases.

places fixed activation functions with learnable univariate functions (e.g., B-splines), directly modeling non-linearities and reducing reliance on redundant linear layers. This makes KAN particularly suitable for capturing hierarchical geometric features in irregular data like point clouds. In terms of parameter efficiency, KAN requires more parameters than MLP for models of the same scale. However, fortunately, smaller-scale KANs exhibit better generalization capabilities. The parameters in KAN are utilized to shape the activation functions, rather than relying on increasing the number of neurons, as in MLP, to enhance model capacity.

Building on this foundation, we propose PointKAN, a point cloud analysis framework that leverages the Kolmogorov-Arnold Network (KAN). Point clouds are first mapped to a high-dimensional space and preprocessed by the Geometric Affine Module (GAM), which performs affine transformations and Softmax pooling (S-Pool) operations. Local features are extracted in parallel by the Local Feature Processing Module, and the aggregated information is passed to the Residual Point (ResP) module to extract global features. To further enhance the computational efficiency of KAN and reduce the number of parameters, the PointKAN-elite version incorporates Efficient-KAN, inspired by Group-Rational KANs (Yang & Wang, 2024). Trained from scratch, PointKAN achieves excellent performance across point cloud datasets (Figure 1(a)), surpassing even pre-trained Point-BERT (Yu et al., 2022) in few-shot learning (Figure 1(b)). It also demonstrates significant reductions in parameters and FLOPs (Figure 1(c-d)), highlighting KAN's potential for 3D point cloud processing.

In summary, this work makes the following contributions.

- We introduce the Kolmogorov-Arnold Network (KAN) for point cloud analysis, named PointKAN, which has a stronger ability to learn complex geometric features.

- We propose PointKAN-elite, an elite version of PointKAN, which significantly reduces both parameter count and computational cost while maintaining comparable experimental results to the original PointKAN.

- During hierarchical point cloud processing, we propose S-Pool to aggregate features within groups, avoiding information loss while providing global context for each group of points.

## 2 RELATED WORK

### 2.1 POINT CLOUD ANALYSIS

The analysis of point clouds has evolved from handcrafted features to deep learning. PointNet (Qi et al., 2017a) pioneered direct processing of unordered points, while PointNet++ (Qi et al., 2017b) introduced hierarchical feature learning. This progress spawned three major paradigms: point-based convolution (Li et al., 2018; Thomas et al., 2019), graph convolution (Wang et al., 2019), and sparse voxelization (Maturana & Scherer, 2015). In recent years, Transformer architectures (Zhao et al., 2021; Guo et al., 2021) have been incorporated to effectively capture long-range dependencies. These models often involve complex local geometric operations that result in high computational overhead, thereby limiting their practical applicability.

PointMLP (Ma et al., 2022) simplifies network architecture to enhance efficiency and performance. It employs a multi-stage design where Farthest Point Sampling (FPS) and K-Nearest Neighbors (KNN) progressively reduce point density at each stage. Local features are extracted via Residual MLP Blocks—comprising linear layers, normalization, and activations with residual connections to mitigate gradient vanishing. Stacking these stages forms a deep hierarchical network for point cloud analysis. The core operation is formulated as $g_i = \Phi_{pos}(\mathcal{A}(\Phi_{pre}(f_{i,j}), |j = 1, \cdots, K))$, where $\Phi_{pre}$ and $\Phi_{pos}$ are residual MLP blocks, $\mathcal{A}$ denotes max-pooling and $f_{i,j}$ represents the feature of the $j$-th point in the $i$-th group, with each group encompassing $K$ points. Despite its simple architecture, PointMLP (Ma et al., 2022) delivers excellent performance on benchmarks like ModelNet40 (Wu et al., 2015) and ScanObjectNN (Uy et al., 2019).

### 2.2 KOLMOGOROV-ARNOLD NETWORK

Kolmogorov-Arnold Network (KAN) (Liu et al., 2024) represent a novel architecture grounded in the Kolmogorov-Arnold Representation Theorem (KART) (Arnold, 2009). Unlike MLP, KAN replaces fixed activation functions with learnable univariate functions. KART establishes that any multivariate continuous function decomposes into finite univariate functions. KAN implements this using B-spline bases and compositional operations, providing a flexible and interpretable framework for high-dimensional function approximation.

KAN demonstrates growing efficacy across domains including computer vision (Ferdaus et al., 2024; Ma et al., 2025; Moradi et al., 2024; Mohan et al., 2024) and medical imaging (Yang et al., 2025; Agrawal et al., 2024). In vision, KAN-augmented convolutional layers replace linear transformations with learnable per-pixel nonlinear activations (Bodner et al., 2024), enhancing accuracy and expressiveness for image recognition. Medical implementations integrate KAN layers into frameworks like U-Net (Ronneberger et al., 2015), improving segmentation accuracy and interpretability (Wu et al., 2024; Li et al., 2024; Tang et al., 2024). For point clouds, PointNet-KAN (Kashefi, 2024) shows better classification and segmentation results versus PointNet (Qi et al., 2017a). However, PointNet's inherent limitations constrain this benchmark, and KAN's potential for complex point cloud tasks remains largely unexplored. Effective integration of KAN with advanced point cloud architectures thus presents a critical research direction.

## 3 METHOD

We propose leveraging KAN for local feature learning in point clouds. To achieve this, we first review the Kolmogorov-Arnold Representation Theorem (KART) and KAN fundamentals(Section 3.1), then detail our model's key designs(Section 3.2) including Efficient-KAN(Section 3.3) in PointKAN-elite.

### 3.1 PRELIMINARIES: KART AND KAN

The Kolmogorov-Arnold Representation Theorem (KART), established by Andrey Kolmogorov and Vladimir Arnold in the 1950s, asserts that any continuous multivariate function $f : [0, 1]^d \to \mathbb{R}$,

$$f(x_1, x_2, \ldots, x_d) = \sum_{q=1}^{2d+1} \phi_q \left( \sum_{p=1}^{d} \psi_{q,p}(x_p) \right), \tag{1}$$

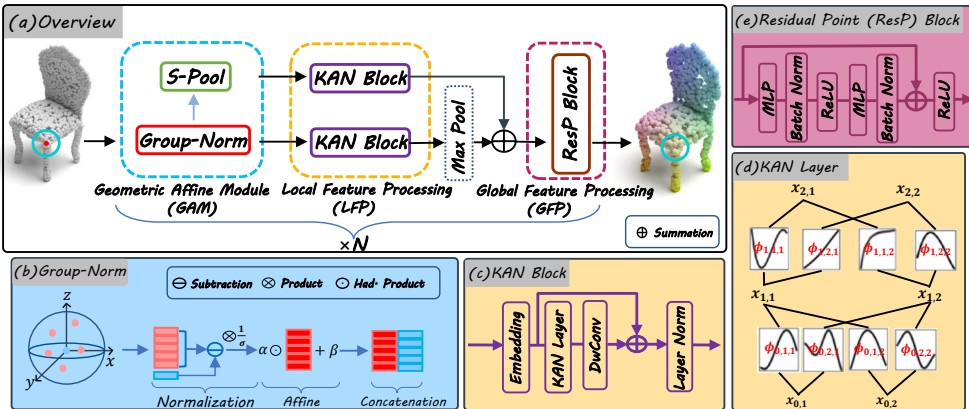

**Figure 2: Illustration of PointKAN**. (a) Each of the four identical stages processes local features through a Geometric Affine Module (GAM), extracts grouped features via Local Feature Processing (LFP), supplements global information after aggregation, and obtains overall features through Global Feature Processing (GFP). This multi-stage repetition progressively enlarges the receptive field for comprehensive geometric capture. (b) Group-Norm comprises normalization, affine transformation, and feature concatenation; (c-d) present KAN Block and Layer details; (e) The Residual Point (ResP) Block integrates MLP with Batch Normalization and ReLU activation.

where $\psi_{q,p} : [0,1] \to \mathbb{R}$ and $\phi_q : \mathbb{R} \to \mathbb{R}$ are univariate continuous functions. Although Equation (1) theoretically enables high-dimensional function representation via two univariate layers, these functions may be highly complex (even fractal) and thus unlearnable by neural networks.

KAN's innovation overcomes KART's two-layer function limitation by implementing a multi-layer network trained via backpropagation. A continuous and smooth univariate spline function $spline(x)$ is generated by the linear combination of B-spline base functions ($B_i(x)$), as shown in Equation (2). The coefficients $c_i$ therein are learnable. The activation function $\phi(x)$ is the sum of $silu(x)$ and $spline(x)$, where $silu(x)$ acts similar to a residual connection.

$$spline(x) = \sum_i c_i B_i(x), \quad \phi(x) = silu(x) + spline(x), \quad (2)$$

by replacing traditional fixed activation functions with this approach, the model's expressiveness and efficiency are enhanced while maintaining its interpretability.

### 3.2 PointKAN

Despite KAN's superior high-dimensional function approximation capabilities and parameter efficiency, its adaptation to 3D point clouds remains challenging. The activation functions in KAN are obtained through a linear combination of basis functions, making them sensitive to input variations and consequently reducing the robustness of the model, while per-dimension parameter storage incurs significant memory overhead for large-scale networks. Additionally, B-spline computations exhibit suboptimal hardware acceleration, leading to inference latency. To overcome these limitations, we propose PointKAN—featuring a geometric affine module and parallel local feature extraction structure—alongside its lightweight variant PointKAN-elite, which reduces memory footprint and accelerate training and inference speed. The complete pipeline is illustrated in Figure 2.

#### 3.2.1 GEOMETRIC AFFINE MODULE

This module integrates Group-Norm and S-Pool for enhanced model robustness. Given input point cloud data $P \in \mathbb{R}^{N \times 3}$, we first apply Farthest Point Sampling (FPS) to select $G$ center points $P_C \in \mathbb{R}^{G \times 3}$, then construct local groups $\{x_p^i \in \mathbb{R}^{K \times 3}\}_{i=1}^{G}$ around each center using K-Nearest Neighbors (KNN). A lightweight PointNet (Qi et al., 2017a) processes each group to generate features $\{f_j^i\}_{j=1}^{K} \in \mathbb{R}^{K \times d}$ (where $d$ denotes feature dimension) and center features $\{f^i\}_{i=1}^{G} \in \mathbb{R}^{G \times d}$, which collectively form the Group-Norm input.

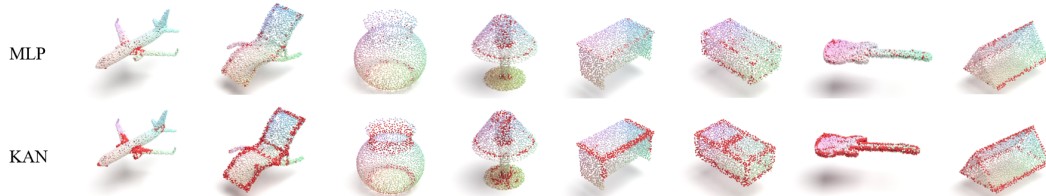

Figure 3: **Local geometric feature sensitivity comparison.** KAN exhibits superior sensitivity to local geometric variations in point clouds compared to MLP.

Group-Norm module performs intra-group feature normalization, learnable affine transformation, and central-point feature propagation. Features $\{f_j^i\}_{j=1}^K$ are normalized:

$$\{\hat{f}_j^i\} = \frac{\{f_j^i\} - f^i}{\sqrt{Var(\{f_j^i\} - f^i) + \epsilon}}, \tag{3}$$

yielding $\mathbb{R}^{K \times d}$ features. We then apply an affine transformation $\tilde{f}_j^i = \alpha \odot \hat{f}_j^i + \beta$ with learnable parameters $\alpha, \beta \in \mathbb{R}^{2d}$ ($\odot$: Hadamard product), followed by concatenation $\tilde{f}_j^i \oplus f^i$ ($\oplus$: feature-dimension concatenation) to propagate central-point features, outputting $\mathbb{R}^{K \times 2d}$ features. Here $\epsilon = 1e^{-5}$ ensures numerical stability, where the affine transformation captures rigid feature transformations while preserving local characteristics.

To address the lack of global context from mutually independent group features when directly feeding Group-Norm outputs to KAN, we introduce S-Pool in the GAM. This aggregates group-wise features, with S-Pool and Group-Norm outputs processed in parallel to supplement global information. Unlike max-pooling which maintains permutation invariance but incurs information loss, our softmax-inspired S-Pool maximizes intra-group feature preservation as defined in Equation (4),

$$\tilde{f}^i = \sum_j^K \frac{\exp \tilde{f}_j^i}{\sum_k^K \exp \tilde{f}_k^i} \cdot \tilde{f}_j^i, \tag{4}$$

where $\tilde{f}^i$ denotes the updated feature of the group center. This S-Pool method proportionally integrates multi-dimensional features of each point within a group into the final center feature, achieving a mapping from $\{\tilde{f}_j^i\}_{j=1,\cdots,K} \in \mathbb{R}^{G \times K \times 2d} \mapsto \tilde{f}^i \in \mathbb{R}^{G \times 2d}$.

### 3.2.2 LOCAL FEATURE PROCESSING

Compared to MLP, KAN employs learnable activation functions, as shown in Equation (2). The key feature of this structure lies in the fact that each spline $B_i(x)$ has a local support region. For a given input $x$, only the basis functions $B_i(x)$ near $x$ are explicitly activated (with larger $c_i$ values), while the contributions of $B_i(x)$ from distant regions can be ignored. This characteristic allows KAN to exhibit excellent sensitivity to local variations and enables adjustments through hyperparameters such as $k$(basis function order) and $G$(grid size) to achieve an appropriate granularity of fitting. The local support property ensures that the model remains focused on the regions around the input $x$, making KAN suitable for structured geometric representations.

To validate that KAN is more suitable for local feature extraction than MLP, we feed point clouds into LFP modules with different architectures and compute the summation of the L2-norm for the output features. Points with values greater than a given threshold are highlighted in red. As depicted in Figure 3, we can intuitively observe that KAN exhibits greater sensitivity to edge points, abrupt change points, and junction points between different parts of the point cloud, demonstrating its superior capability in learning local feature variations.

The LFP structure is depicted in Figure 2(c). Let $\mathcal{F}(\cdot)$ represent the processing procedure of the KAN block, which is expressed as:

$$\mathcal{F}(f_{in}) = f_{in} + DwConv(\Phi(f_{in})), \tag{5}$$

where $f_{in}$ denotes the input point cloud features, $\Phi$ represents the KAN layer, and DwConv is a depthwise convolution, which assists KAN in learning rich feature representations from high-dimensional information.

### 3.2.3 GLOBAL FEATURE PROCESSING

The GFP structure, as shown in Figure 2(e), fully leverages the ability of MLP to achieve cross-channel fusion within fully connected layers—MLP focuses on reweighting features across channels during aggregation, rather than directly adding the results of activation function processing as in KAN.(More detailed explanations are provided in the appendix) This structure satisfies the invariance requirements of point cloud processing, and the purely feed-forward operations ensure computational efficiency in large-scale data processing.

In general, instead of using complex local geometric extraction structures, one stage of PointKAN consists of three components: the Geometric Affine Module, the Local Feature Processing, and the Global Feature Processing. We construct a deep network for hierarchical point-cloud processing through four recursively repeated stages.

### 3.3 EFFICIENT-KAN

KAN's B-spline activation functions $\phi(x)$ exhibit computational inefficiency due to recursive computation and poor GPU parallelization. Each input-output pair requires dedicated parameters and base functions, causing exponential parameter growth with hidden layer width. To mitigate these limitations, we introduce Efficient-KAN in PointKAN-elite as a scalable alternative.

In comparison to KAN, Efficient-KAN utilize rational functions as the base functions within KAN, replacing the B-spline functions. Specifically, each activation function $\phi(x)$ is computed using rational polynomials $P(x)$ and $Q(x)$ as defined in Equation (6), where the degrees of $P(x)$ and $Q(x)$ are denoted as $m$ and $n$, respectively.

$$\phi(x) = wF(x) = w\frac{P(x)}{Q(x)} = w\frac{a_0 + a_1 x + \cdots + a_m x^m}{\sqrt{1 + (b_1 x + \cdots + b_n x^n)^2}}, \tag{6}$$

where $\{a_i\}_{i=1,\cdots,m}$ and $\{b_j\}_{j=1,\cdots,n}$ are coefficients of the rational functions, and $w$ is the scaling factor. These parameters are trained via backpropagation, where rational functions implemented with CUDA demonstrate lower computational complexity than B-spline functions and enhanced parallelism. We specifically implement two corresponding optimizations.

First, we employed Horner's method (Horner, 1815) for polynomial evaluation to further reduce computational overhead. Horner's method is described as follows:

$$a_0 + a_1 x + \cdots + a_m x^m = a_0 + x\left(a_1 + x\left(a_2 + x(\cdots)\right)\right), \tag{7}$$

which evaluates a polynomial of degree $m$ using only $m$ multiplications and $m$ additions. Subsequently, the explicit gradients of $F(x)$ with respect to $\frac{\partial F}{\partial a_m}$, $\frac{\partial F}{\partial b_n}$ and $\frac{\partial F}{\partial x}$ can be computed respectively:

$$\frac{\partial F}{\partial a_m} = \frac{x^m}{Q(x)}, \quad \frac{\partial F}{\partial b_n} = -x^n\frac{G(x)}{Q^3(x)}P(x), \quad \frac{\partial F}{\partial x} = \frac{\partial P(x)}{\partial x}\frac{1}{Q(x)} - \frac{\partial Q(x)}{\partial x}\frac{P(x)}{Q^2(x)} \tag{8}$$

which $G(x) = b_1 x + \cdots + b_n x^n$, $\frac{\partial P(x)}{\partial x} = a_1 + 2a_2 x + \cdots + ma_m x^{m-1}$ and $\frac{\partial Q(x)}{\partial x} = \frac{G(x)}{Q(x)}\left(b_1 + 2b_2 x + \cdots + nb_n x^{n-1}\right)$. All factors in the result can be further accelerated computationally using Horner's method.

Another significant change in Efficient-KAN is the grouping of input channels, which allows parameter sharing within groups to reduce the number of parameters and computational load. As illustrated in Figure 4, MLP utilizes fixed activation functions, while KAN assigns an activation function to each input-output pair. In contrast, Efficient-KAN integrates the approaches of the former two by sharing activa-

Table 1: **The parameter comparison among single-layer MLP, Vanilla KAN, and Efficient-KAN.**

| Model | Parameter count |
|---|---|
| MLP | $d_{\text{in}} \times d_{\text{out}} + d_{\text{out}}$ |
| Vanilla KAN | $d_{\text{in}} \times d_{\text{out}} \times (G + k + 2) + d_{\text{out}}$ |
| Efficient-KAN (Ours) | $d_{\text{in}} \times d_{\text{out}} + d_{\text{out}} + (n + m \times g)$ |

tion functions within groups to minimize the number of parameters. Table 1 presents a quantitative comparison of the parameter counts among the single-layer MLP, the Vanilla KAN, and Efficient-KAN, in which $d_{in}$ and $d_{out}$ represent the input and output dimensions, respectively; $k$ denotes

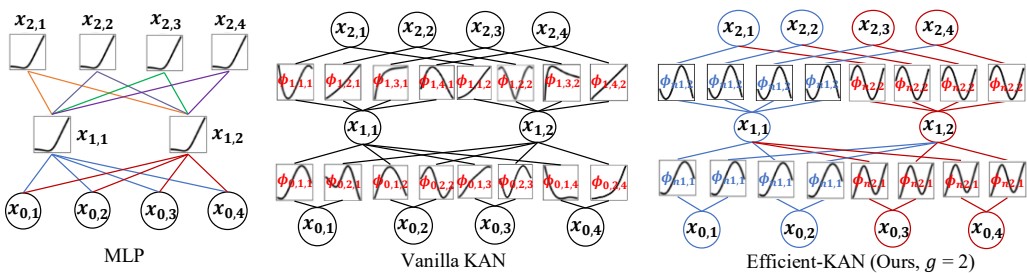

Figure 4: **Comparing among MLP, Vanilla KAN, and Efficient-KAN.** MLP: each output channel uses the same activation function. Vanilla KAN: each input-output pair has distinct activation functions. Efficient-KAN: input-output pairs within the same group employ the same activation function.

Table 2: **Classification results on ModelNet40 and ScanObjectNN**. The model parameters Param.(M), FLOPs(G), overall accuracy(%) and Train/Test speed(samples/s) are reported. We compare the KAN-based model with methods based on CNN, GCN, Transformer, Mamba, and MLP.

| Method | Param.(M) ↓ | FLOPs(G) ↓ | ModelNet40 | ScanObjectNN | | | Train speed | Test speed |
| | | | 1k P ↑ | OBJ_BG ↑ | OBJ_ONLY ↑ | PB_T50_RS ↑ | (samples/s) ↑ | (samples/s) ↑ |
|---|---|---|---|---|---|---|---|---|
| *Based on CNN or GCN* | | | | | | | | |
| DGCNN (Wang et al., 2019) | 1.8 | 2.4 | 92.9 | 82.8 | 86.2 | 78.1 | - | - |
| PointCNN (Li et al., 2018) | **0.6** | - | 92.5 | 86.1 | 85.5 | 78.5 | - | - |
| PointConv (Wu et al., 2019) | 18.6 | - | 92.5 | - | - | - | 17.9 | 10.2 |
| KPConv (Wu et al., 2019) | 15.2 | - | 92.9 | - | - | - | 31.0 | 80.0 |
| GBNet (Qiu et al., 2021b) | 8.8 | - | 93.3 | - | - | 81.0 | - | - |
| *Based on Transformer or Mamba* | | | | | | | | |
| Transformer (Vaswani et al., 2017) | 22.1 | 4.8 | 91.4 | 79.9 | 80.6 | 77.2 | - | - |
| PCT Guo et al. (2021) | 2.9 | 2.3 | 93.2 | - | - | - | - | - |
| PointMamba (Liang et al., 2024) | 12.3 | 3.6 | - | **88.3** | **87.8** | 82.5 | - | - |
| PCM (Zhang et al., 2024) | 34.2 | 45.0 | 93.4±0.2 | - | - | **88.1±0.3** | - | - |
| *Based on MLP* | | | | | | | | |
| PointNet (Qi et al., 2017a) | 3.5 | **0.5** | 89.2 | 73.3 | 79.2 | 68.0 | - | - |
| PointNet++ (Qi et al., 2017b) | 1.5 | 1.7 | 90.7 | 82.3 | 84.3 | 77.9 | **223.8** | **308.5** |
| DRNet (Qiu et al., 2021a) | - | - | 93.1 | - | - | 80.3 | - | - |
| SimpleView (Goyal et al., 2021) | - | - | 93.4 | - | - | 80.5±0.3 | - | - |
| PRA-Net (Cheng et al., 2021) | | 2.3 | **93.7** | - | - | 81.0 | - | - |
| PointMLP (Ma et al., 2022) | 12.6 | 31.4 | 93.3 | - | - | 85.0±0.3 | 47.1 | 112 |
| *Based on KAN* | | | | | | | | |
| PointNet-KAN (Kashefi, 2024) | - | - | 90.5 | - | - | 60.2 | - | - |
| *PointKAN (Ours)* | 10.0 | 9.1 | **93.7** | 87.1 | 86.4 | 83.5 | 41.8 | 91.4 |
| *PointKAN-elite (Ours)* | 3.1 | 2.3 | 93.3 | 84.3 | 85.3 | 84.1 | 44.7 | 154.3 |

the spline order; $G$ indicates the grid size; $m$ and $n$ correspond to the polynomial degrees of rational functions; and $g$ is the number of groups. The parameter count of Efficient-KAN introduces only a constant term compared to MLP, whereas original KAN's parameter scale grows roughly as $(G+k+2)$ times that of MLP.

# 4 EXPERIMENTS

## 4.1 IMPLEMENTATION DETAILS

Experiments are conducted on three benchmarks using NVIDIA RTX 4090: ModelNet40 (Wu et al., 2015) and ScanObjectNN (Uy et al., 2019) for classification, and ShapeNetPart (Yi et al., 2016) for part segmentation, with hyperparameters aligned to PointMLP (Ma et al., 2022) for fair comparison. The PointKAN architecture employs four identical stages per task, each containing a GAM, LFP, and GFP module, using a batch size of 18. For classification, SGD optimizes inputs of 1,024 points grouped into 24 per partition; for segmentation, Adam with cosine decay handles 2,048 points (including normals) grouped into 32 per partition. The PointKAN-elite variant increases the batch size to 32 while halving learning rates, retaining other configurations. All ablation studies default to PointKAN and are evaluated primarily on ModelNet40.

## 4.2 EXPERIMENTAL RESULTS ON DOWNSTREAM TASKS

PointKAN (including PointKAN-elite) demonstrates inherent advantages over PointMLP (Ma et al., 2022) in multi-dimensional benchmarking across downstream tasks. Our models are trained from

scratch for each specific task, with no implementation of pre-training methodologies or ensemble voting mechanisms in any experimental configuration.

**Synthetic Object Classification on ModelNet40.** The ModelNet40 (Wu et al., 2015) dataset constitutes a comprehensive and clean collection of 3D CAD models, encompassing 12,311 human-annotated models spanning 40 distinct categories. The dataset is partitioned into 9,843 models for training and 2,468 models for testing. As demonstrated in Table 2, we report overall accuracy (OA) on the test set. Among these methods, our PointKAN achieves a 0.4% enhancement in OA (93.7% vs. 93.3%) compared to PointMLP (Ma et al., 2022) when using 1k points. Notably, PointKAN reduces parameter count by 21% (10M vs. 12.6M) and computational complexity (FLOPs) by 71% (9.1G vs. 31.4G) relative to PointMLP. Moreover, PointKAN-elite further reduces the number of parameters (3.1M) and computational cost (2.3G FLOPs) while maintaining accuracy. Additionally, its inference speed surpasses that of PointMLP (154.3 samples/s vs. 112 samples/s).

**Real-world Object Classification on ScanObjectNN.** The ScanObjectNN dataset (Uy et al., 2019), derived from 3D scans of real-world indoor scenes, comprises 15,000 point cloud object samples spanning 15 daily object categories (e.g., chairs, tables, monitors) and encompassing 2,902 distinct object instances. As shown in Table 2, we conducted experiments on three progressively challenging variants of ScanObjectNN: OBJ-BG (objects with background), OBJ-ONLY (isolated objects), and PB-T50-RS (perturbed objects with real-world noise). Among models trained from scratch, PointKAN achieved exceptional overall accuracy (OA) with fewer training epochs (200 epochs), attaining 87.1% on OBJ-BG

Table 3: **Few-shot classification on ModelNet40**. Overall accuracy (%) without voting and pre-training strategies is reported.

| Method | 5-way | | 10-way | |
|---|---|---|---|---|
| | 10-shot ↑ | 20-shot ↑ | 10-shot ↑ | 20-shot ↑ |
| *Based on CNN* | | | | |
| DGCNN (Wang et al., 2019) | 31.6 ± 2.8 | 40.8 ± 4.6 | 19.9 ± 2.1 | 16.9 ± 1.5 |
| DGCNN+*OcCo (Wang et al., 2021)* | 90.6 ± 2.8 | 92.5 ± 1.9 | 82.9 ± 1.3 | 86.5 ± 2.2 |
| DGCNN-CrossPoint (Afham et al., 2022) | 92.5 ± 3.0 | 94.9 ± 2.1 | 83.6 ± 5.3 | 87.9 ± 4.2 |
| *Based on Transformer or Mamba* | | | | |
| Transformer (Vaswani et al., 2017) | 87.8 ± 5.2 | 93.3 ± 4.3 | 84.6 ± 5.5 | 89.4 ± 6.3 |
| MAMBA3D (Han et al., 2024) | 92.6 ± 3.7 | 96.9 ± 2.4 | 88.1 ± 5.3 | 93.1 ± 3.6 |
| Point-BERT (Yu et al., 2022) | 94.6 ± 3.1 | 96.3 ± 2.7 | 91.0 ± 5.4 | 92.7 ± 5.1 |
| MaskPoint (Liu et al., 2022) | 95.0 ± 3.7 | 97.2 ± 1.7 | **91.4 ± 4.0** | 93.4 ± 3.5 |
| *Based on MLP* | | | | |
| PointNet (Qi et al., 2017a) | 52.0 ± 3.8 | 57.8 ± 4.9 | 46.6 ± 4.3 | 35.2 ± 4.8 |
| PointNet-CrossPoint (Afham et al., 2022) | 90.9 ± 1.9 | 93.5 ± 4.4 | 86.4 ± 4.7 | 90.2 ± 2.2 |
| PointMLP (Ma et al., 2022) | 90.5 ± 4.2 | 94.2 ± 4.1 | 84.1 ± 5.6 | 91.5 ± 5.1 |
| *Based on KAN* | | | | |
| *PointKAN(Ours)* | **95.9 ± 3.1** | **97.9 ± 2.0** | 90.5 ± 4.9 | **94.0 ± 3.5** |
| *PointKAN-elite(Ours)* | 95.2 ± 3.1 | 96.3 ± 3.0 | 88.0 ± 5.1 | 93.5 ± 4.0 |

and 86.4% on OBJ-ONLY. However, on the most complex PB-T50-RS variant, PointKAN slightly underperformed PointMLP (83.5% vs. 84.7% OA). Notably, PointKAN-elite demonstrated a 0.6% OA improvement over PointKAN (84.1% vs. 83.5%) on PB-T50-RS while achieving a significant reduction in parameters and FLOPs.

**Few-shot Learning.** To validate the few-shot transfer capability of PointKAN, we conduct few-shot classification experiments following prior work. Adopting the "n-way, m-shot" protocol on ModelNet40, where $n \in \{5, 10\}$ denotes the number of randomly selected object categories and $m \in \{10, 20\}$ represents the number of sampled instances per category, we restrict model training to $n \times m$ samples exclusively. During testing, 20 novel instances per class are randomly selected as evaluation data. For each configuration, we perform 10 independent trials and report the mean accuracy with standard deviation. As shown in Table 3, PointKAN achieves superior accu-

Table 4: **Part segmentation results on ShapeNet-Part.** The class mIoU (Cls. mIoU) and the instance mIoU (Inst. mIoU) are reported.

| Method | Cls. mIoU↑ | Inst. mIoU↑ | Param.(M)↓ | FLOPs(G)↓ |
|---|---|---|---|---|
| *Based on CNN or GCN* | | | | |
| PCNN (Atzmon et al., 2018) | 81.8 | 85.1 | 5.4 | - |
| DGCNN (Wang et al., 2019) | 82.3 | 85.2 | 1.3 | 12.4 |
| PointCNN (Li et al., 2018) | **84.6** | 86.1 | - | - |
| RS-CNN (Liu et al., 2019) | 84.0 | **86.2** | - | - |
| SpiderCNN (Xu et al., 2018) | 82.4 | 85.3 | - | - |
| *Based on MLP* | | | | |
| PointNet (Qi et al., 2017a) | 80.4 | 83.7 | 3.6 | 4.9 |
| PointNet++ (Qi et al., 2017b) | 81.9 | 85.1 | **1.0** | 4.9 |
| PointMLP (Ma et al., 2022) | **84.6** | 86.1 | 16.0 | 5.8 |
| *Based on KAN* | | | | |
| *PointKAN(Ours)* | 83.9 | 85.6 | 13.8 | 3.8 |
| *PointKAN-elite(Ours)* | 84.3 | 86.0 | 6.2 | **1.4** |

racy compared to CNN-based variants and even surpasses Transformer/Mamba-based counterparts like Point-BERT (Yu et al., 2022) and MAMBA3D (Han et al., 2024). The only exception occurs in the 10-way 10-shot setting, where PointKAN marginally underperforms MaskPoint (Liu et al., 2022) and Point-BERT. In comparisons between KANs and MLPs, PointKAN demonstrates consistent overall accuracy (OA) gains over PointMLP: +5.4%, +3.7%, +6.4%, and +2.5% across configurations. Notably, despite being trained from scratch, PointKAN (and its lightweight vari-

Table 5: **Impact of components on model robustness.** We conduct ablation studies on ModelNet40 and the overall accuracy (%) are reported.

| Affine | S-Pool | Translation and scaling | Point dropout | Noise |
|--------|--------|------------------------|---------------|-------|
| ✗ | ✓ | 92.5 | 91.0 | 90.3 |
| ✓ | ✗ | 93.0 | 91.6 | 90.5 |
| ✗ | ✗ | 90.7 | 90.2 | 90.0 |
| ✓ | ✓ | **93.7** | **92.9** | **91.8** |

Table 6: **Ablation on KANs' depth and hidden layer width.** $d_{in}$ denotes the input feature dimension. The mean accuracy (%) and overall accuracy (%) are reported.

| Depth | Width | mAcc(%) | OA(%) |
|-------|-------|---------|-------|
| 3 layers | $d_{in} \times 1/3$ | 90.6 | 93.0 |
| | $d_{in} \times 1/2$ | **91.4** | **93.7** |
| | $d_{in} \times 2/3$ | 90.5 | 92.8 |
| | $d_{in}$ | 90.4 | 93.5 |
| 4 layers | $d_{in} \times 1/3$ | 90.1 | 92.9 |
| | $d_{in} \times 1/2$ | 91.0 | 93.2 |

ant PointKAN-elite) outperforms Point-BERT (which employs self-supervised pre-training) in most tasks, underscoring its exceptional generalization capacity and knowledge transfer ability.

**Part Segmentation on ShapeNetPart.** We conduct part segmentation experiments on ShapeNetPart (Yi et al., 2016) to predict more fine-grained category labels for each point within the samples. The ShapeNetPart dataset comprises 16,881 3D instances spanning 16 object categories with a total of 50 part labels. As illustrated in Table 4, we systematically compared our method against representative approaches based on CNNs, GCNs, and MLPs, reporting both the mean Intersection over Union (mIoU) across all categories (Cls.) and instances (Inst.). Due to space limitations, we report the mIoU for each category in the supplementary materials. The experimental results demonstrate that our PointKAN (including PointKAN-elite variant) achieves highly competitive performance compared to PointMLP while exhibiting significant reductions in both parameter count (13.8M vs. 16.0M) and FLOPs(3.8G vs. 5.8G). PointKAN-elite demonstrates even more significant improvements(6.2M vs. 16.0M, 1.4G vs. 5.8G). (Visualization of part segmentation in the appendix.)

### 4.3 ABLATION STUDY

**Architecture Ablation for Model Robustness Validation.** Ablation studies demonstrate that the affine transformation and the S-Pool structure in the GAM exhibit strong robustness against various point cloud data perturbations. The affine operation enhances model stability by extracting features from diverse geometric structures across different local regions. The S-Pool structure compensates for global information loss during local grouping processes through parallel processing with GroupNorm. Results (Table 5) presented that under point cloud perturbations including translation and scaling, random point dropping, and noise, the OA reaches 93.7, 92.9, and 91.8, respectively.

**Ablation of Built-in Parameters in KAN.** KAN critically determines model performance. Firstly, we analyze two parameters, $k$ (the order of the B-spline function) and $G$ (grid size) in Figure 5. Higher $k$ enhances curve continuity while $G$ controls knot interval density, jointly affecting flexibility. However, excessive values cause overfitting and computational complexity.

In addition, guided by KAN's generalization principles (Liu et al., 2024), we balance network scale: insufficient capacity impairs feature learning, while oversized networks harm generalization.

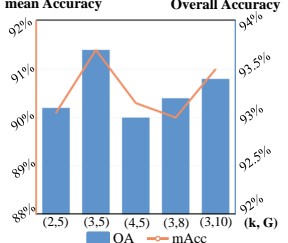

Figure 5: **Ablation of Built-in Parameters** $k$ **and** $G$ **in KANs.** The mean accuracy (%) and overall accuracy (%) are reported.

Through architectural experiments (Table 6), a three-layer KAN with $k = 3$, $G = 5$, and hidden dimension $d_{hid} = d_{in}/2$ achieves optimal accuracy.

### 5 CONCLUSION

We propose PointKAN, an efficient point cloud analysis architecture using KAN. PointKAN outperforms the MLP-based architecture PointMLP across multiple tasks, demonstrating KAN's powerful capability in extracting local detailed features. Additionally, a more efficient lightweight version, PointKAN-elite, is proposed, which further reduces the number of parameters and computational cost while maintaining accuracy. We expect that PointKAN can promote the application of KAN in point cloud analysis and fully leverage the unique advantages of KAN over MLP.

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

## A    REPRODUCIBILITY STATEMENT

We have already elaborated on all the models or algorithms proposed, experimental configurations, and benchmarks used in the experiments in the main body or appendix of this paper. Furthermore, we declare that the entire code used in this work will be released after acceptance.

## B    THE USE OF LARGE LANGUAGE MODELS

We use large language models solely for polishing our writing, and we have conducted a careful check, taking full responsibility for all content in this work.

## C    VISUALIZATION OF PART SEGMENTATION

We provide a comparative visualization between the segmentation ground truths and predicted outcomes in Figure 6. Notably, the predictions from PointKAN closely align with the annotated ground truth labels. These compelling findings substantiate the considerable potential of KAN for point cloud analysis tasks.

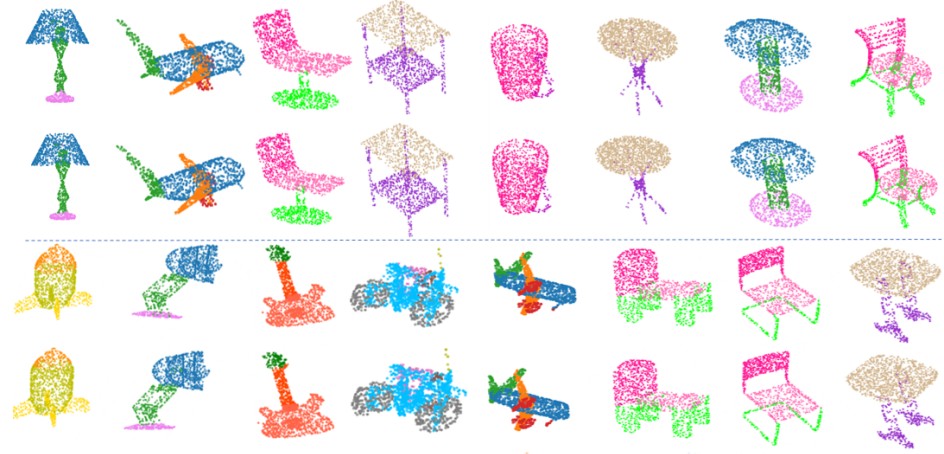

Figure 6: **Part segmentation results on ShapeNetPart.** Top line is ground truth and bottom line is our prediction.

## D    MORE DETAILED ABLATION STUDIES

We perform comprehensive ablation studies on the architecture and KAN parameters. All results are obtained from models trained from scratch on ModelNet40.

### D.1    ARCHITECTURE ABLATION

We investigate the effectiveness of each component proposed in PointKAN, and the results are presented in Table 7. From this, we can draw the following findings: First, the GAM increases the overall accuracy of the model by 3.0%. This is reasonable because in the subsequent LFP module, the outputs of S-Pool and Group-Norm are processed in parallel, which compensates for the lack of global information when dealing with local groupings of point clouds. Affine transformations also enhance the model's robustness. Second, removing the LFP module and directly performing a max-pooling operation on the output of Group-Norm and adding it to the result of S-Pool leads to a 1.0% decrease in OA after the LFP module. This demonstrates the powerful local feature extraction ability of KAN and also validates the effectiveness of the LFP module for unordered point cloud data. Finally, the Residual Point Block in the GFP module is responsible for processing the entire point cloud to extract the final global features. Removing it reduces the overall accuracy by 0.5%. Combining all these components together, we achieved the best result with an overall accuracy of 93.7%.

Table 7: **Component ablation studies on ModelNet40 test set.** The mean accuracy (%) and overall accuracy (%) are reported.

| GAM | LFP | GFP | mAcc(%) | OA(%) |
|:---:|:---:|:---:|:---:|:---:|
| ✗ | ✓ | ✓ | 88.5 | 90.7 |
| ✓ | ✗ | ✓ | 89.9 | 92.7 |
| ✓ | ✓ | ✗ | 90.7 | 93.2 |
| ✓ | ✓ | ✓ | **91.4** | **93.7** |

## D.2 DEPTH-WISE CONVOLUTION

Depthwise separable convolution Chollet (2017) (DwConv), a computationally efficient operation widely implemented in lightweight neural networks, substantially reduces both computational overhead and parameter count while preserving model performance relative to traditional convolution. This operation, when integrated with KANs, facilitates cross-channel information fusion in high-dimensional channels to generate advanced semantic features. Our ablation studies evaluating the impact of DwConv reveal notable performance gains, with Table 8 showing a 1.2% improvement in mean accuracy (91.4% vs. 90.2%) and a 1.0% increase in overall accuracy (93.7% vs. 92.7%).

Table 8: Component ablation studies on ModelNet40 test set. The mean accuracy (%) and overall accuracy (%) are reported.

| DwConv | mAcc(%) | OA(%) |
|:---:|:---:|:---:|
| ✗ | 90.2 | 92.7 |
| ✓ | **91.4** | **93.7** |

## D.3 THE FRAMEWORK OF GLOBAL FEATURE PROCESSING (GFP)

We conduct ablation studies on the GFP framework using KAN Block and Resp Block respectively, with results shown in Table 9. The MLP-based Resp Block demonstrates superior performance in global deep feature fusion. Below, we analyze the mathematical principles underlying both approaches.

In point cloud analysis models, KAN is better suited for local feature extraction while MLP excels at global feature aggregation. MLP employs a fully-connected structure where each neuron computes:

$$\mathbf{y} = \sigma\left(\mathbf{W}\mathbf{x} + \mathbf{b}\right), \tag{9}$$

where $\mathbf{x} = (x_1, x_2, \cdots, x_n)$ represents the input vector, $\mathbf{W} \in \mathbb{R}^{m \times n}$ is the weight matrix of the hidden layer, and $\mathbf{b} \in \mathbb{R}^m$ represents the bias vector. $\sigma(\cdot)$ applies the same activation function (such as ReLU, Sigmoid, etc.) to each element of the vector respectively. This global connectivity entails that each output neuron depends on all input neurons, while the predetermined activation function exhibits limited sensitivity to local variations. Although ReLU introduces nonlinearity, its simplistic global activation mechanism struggles to capture subtle local geometric variations arising from complex point interactions in point clouds.

Conversely, KAN – based on the Kolmogorov-Arnold Representation Theorem (KART) – replaces weight matrices with learnable univariate functions $\phi_{ij}$ parameterized by B-splines:

$$\phi_{ij}(x) = \sum_k c_{ij}^k B^k(x), \tag{10}$$

where $c_{ij}$ is learnable coefficient and $B(x)$ represents B-spline base function. Network computes node outputs as $y_j = \sum_i \phi_{ij}(x_i)$. Critically, each $\phi_{ij}(x)$ has a local support domain: Only B-spline basis functions $B^k(x)$ near a specific input $x_i$ are significantly activated, with minimal contributions from distant regions. This property confers exceptional sensitivity to local variations and enables fine-grained fitting. For a point $p_i$ in point cloud, features from its local neighborhood $\mathcal{N}_i$ serve as input $x_i$. The function $\phi_{ij}(x)$ learns a highly refined nonlinear transformation of $\mathcal{N}_i$, with controllable granularity via hyperparameters $k$(spline order) and $G$(grid size). The local support characteristic ensures focused attention on patterns within the vicinity of $x_i$, making KAN inherently suitable for characterizing local geometric structures.

Table 9: Ablation studies on the framework of Global Feature Processing (GFP). The mean accuracy (%) and overall accuracy (%) are reported.

| The framework of GFP | mAcc(%) | OA(%) |
|---|---|---|
| KAN Block | 90.0 | 93.2 |
| Resp Block | **91.4** | **93.7** |

Table 10: Part segmentation results on ShapeNetPart dataset. The class mIoU (Cls. mIoU) and the instance mIoU (Inst. mIoU) are reported.

| Method | Cls. mIoU | Inst. mIoU | aero | bag | cap | car | chair | earp-hone | guitar | knife | lamp | laptop | motor-bike | mug | pistol | rocket | skate-board | table |
|---|---|---|---|---|---|---|---|---|---|---|---|---|---|---|---|---|---|---|
| *Based on CNNs or GNNs* | | | | | | | | | | | | | | | | | | |
| PCNN (Atzmon et al., 2018) | 81.8 | 85.1 | 82.4 | 80.1 | 85.5 | 79.5 | 90.8 | 73.2 | 91.3 | 86.0 | 85.0 | 95.7 | 73.2 | 94.8 | 83.3 | 51.0 | 75.0 | 81.8 |
| DGCNN (Wang et al., 2019) | 82.3 | 85.2 | 84.0 | 83.4 | 86.7 | 77.8 | 90.6 | 74.7 | 91.2 | 87.5 | 82.8 | 95.7 | 66.3 | 94.9 | 81.1 | 63.5 | 74.5 | 82.6 |
| PointCNN (Li et al., 2018) | 84.6 | 86.1 | 84.1 | 86.5 | 86.0 | 80.8 | 90.6 | 79.7 | 92.3 | 88.4 | 85.3 | 96.1 | 77.2 | 95.2 | 84.2 | 64.2 | 80.0 | 83.0 |
| RS-CNN (Liu et al., 2019) | 84.0 | 86.2 | 83.5 | 84.8 | 88.8 | 79.6 | 91.2 | 81.1 | 91.6 | 88.4 | 86.0 | 96.0 | 73.7 | 94.1 | 83.4 | 60.5 | 77.7 | 83.6 |
| SpiderCNN (Xu et al., 2018) | 82.4 | 85.3 | 83.5 | 81.0 | 87.2 | 77.5 | 90.7 | 76.8 | 91.1 | 87.3 | 83.3 | 95.8 | 70.2 | 93.5 | 82.7 | 59.7 | 75.8 | 82.8 |
| KPConv (Thomas et al., 2019) | 85.1 | 86.4 | 84.6 | 86.3 | 87.2 | 81.1 | 91.1 | 77.8 | 92.6 | 88.4 | 82.7 | 96.2 | 78.1 | 95.8 | 85.4 | 69.0 | 82.0 | 83.6 |
| *Based on MLPs* | | | | | | | | | | | | | | | | | | |
| PointNet (Qi et al., 2017a) | 80.4 | 83.7 | 83.4 | 78.7 | 82.5 | 74.9 | 89.6 | 73.0 | 91.5 | 85.9 | 80.8 | 95.3 | 65.2 | 93.0 | 81.2 | 57.9 | 72.8 | 80.6 |
| PointNet++ (Qi et al., 2017b) | 81.9 | 85.1 | 82.4 | 79.0 | 87.7 | 77.3 | 90.8 | 71.8 | 91.0 | 85.9 | 83.7 | 95.3 | 71.6 | 94.1 | 81.3 | 58.7 | 76.4 | 82.6 |
| PointMLP (Ma et al., 2022) | 84.6 | 86.1 | 83.5 | 83.4 | 87.5 | 80.5 | 90.3 | 78.2 | 92.2 | 88.1 | 82.6 | 96.2 | 77.5 | 95.8 | 85.4 | 64.6 | 83.3 | 84.3 |
| *Based on KANs* | | | | | | | | | | | | | | | | | | |
| PointNet-KAN (Kashefi, 2024) | - | 83.3 | 81.0 | 76.8 | 79.8 | 74.6 | 88.7 | 65.4 | 90.9 | 85.3 | 79.9 | 95.0 | 65.3 | 93.0 | 83.0 | 54.3 | 71.9 | 81.6 |
| *PointKAN(Ours)* | 83.9 | 85.6 | 83.3 | 83.0 | 87.2 | 78.6 | 89.4 | 81.2 | 91.6 | 87.1 | 82.7 | 95.9 | 75.6 | 94.4 | 84.0 | 64.7 | 81.5 | 82.9 |
| *PointKAN-elite(Ours)* | 84.3 | 86.0 | 83.3 | 83.4 | 85.0 | 79.7 | 90.4 | 80.7 | 91.9 | 87.9 | 82.5 | 96.0 | 78.2 | 94.9 | 84.9 | 62.8 | 83.2 | 83.8 |

# E  PART SEGMENTATION: mIOU FOR EACH CATEGORY

As shown in Table 10, our comprehensive evaluation of part segmentation performance demonstrates that our approach achieves results comparable to PointMLP Ma et al. (2022) for most categories, with superior performance particularly evident in earphone, lamp, and rocket classifications. Most notably, both PointKAN and PointKAN-elite variants achieve these results while significantly reducing the number of parameters and computational overhead.

