# OpenReview forum: "KAN or MLP? Point Cloud Shows the Way Forward"
_ICLR.cc/2026/Conference — ICLR 2026 Conference Withdrawn Submission_

### Official Review · Reviewer_pEJH · 2025-10-17

**Soundness:** 3
**Presentation:** 2
**Contribution:** 3
**Rating:** 6
**Confidence:** 5

**Summary:**

This paper proposes PointKAN, introducing the Kolmogorov-Arnold Network (KAN) into the framework of point cloud analysis. This framework employs a hierarchical structure to progressively expand the receptive field. Each layer comprises three major components: the Geometric Affine Module (GAM), Local Feature Processing (LFP), and Global Feature Processing (GFP). The LFP module enhances local feature extraction through the KAN architecture. The GAM combines Group-Norm and S-Pool to improve robustness and supplement global information. The GFP utilizes an MLP for cross-channel fusion. To further optimize efficiency, the paper introduces a lightweight variant, PointKAN-elite, which reduces computational overhead through Efficient-KAN techniques (rational function substitution for B-splines and channel-group parameter sharing). Experiments validated performance across three benchmarks: ModelNet40, ScanObjectNN, and ShapeNetPart. PointKAN not only outperforms PointMLP in classification and segmentation tasks but also achieves significantly reduced computational overhead when processing large-scale point clouds. In few-shot learning tasks, its generalization capability surpasses PointMLP, confirming KAN's unique advantages in point cloud analysis.

**Strengths:**

1. The paper accurately identifies the inherent limitations of MLP in point cloud processing: fixed activation functions (such as ReLU) exhibit insufficient sensitivity to local geometric variations, necessitating deeper networks to enhance expressive power—which leads to parameter redundancy and computational inefficiency. This problem identification aligns with the fundamental “irregular and sparse” nature of point clouds and closely matches the practical application bottlenecks of existing MLP-based methods (e.g., PointMLP), demonstrating clear real-world relevance for the research motivation.

2. Pioneering the application of the KAN system to point cloud analysis: Breaking free from the constraints of MLP's fixed activation functions, this approach leverages KAN's “learnable univariate functions (B-splines)” to enhance sensitivity to local geometric features (e.g., edge points, abrupt change points) (visualized in Figure 3). This establishes a novel paradigm for point cloud feature extraction, distinct from existing CNN, Transformer, or Mamba-dominated point cloud architectures.

3. Efficient-KAN optimization with targeted improvements: Addressing native KAN's issues of “low B-spline computation efficiency and parameter growth that scales exponentially with dimension,” we propose “rational function substitution for B-splines + channel grouping with parameter sharing.” This significantly reduces PointKAN-elite's parameters (6.2M) and FLOPs (1.4G) compared to PointMLP (16.0M/5.8G) while maintaining comparable accuracy, achieving a balanced trade-off between "performance - efficiency" balance.

4. Comprehensive benchmark coverage: Covers three task categories—synthetic point clouds (ModelNet40), real-world point clouds (ScanObjectNN), and fine-grained segmentation (ShapeNetPart)—including few-shot learning scenarios. This holistically validates model performance across “standard tasks - extreme data - generalization transfer.”

5. Rigorous comparison and ablation: Conducts in-depth comparisons with MLP-based methods (PointNet/PointMLP) while benchmarking against mainstream architectures like CNN/GCN (DGCNN/PointCNN) and Transformer/Mamba (Point-BERT/PointMamba). Ablation experiments cover core components (GAM/LFP/GFP), KAN parameters (B-spline degree k, grid size G), and Efficient-KAN optimization strategies (Tables 5–8), clearly demonstrating the effectiveness of each design.

**Weaknesses:**

1. The paper primarily compares PointMLP with traditional CNN/Transformer architectures but fails to conduct a comprehensive performance and efficiency comparison with recent efficient point cloud architectures such as PointMamba (Liang et al., 2024) and PointRWKV (He et al., 2024). These methods also emphasize “efficient long-sequence modeling” as their core advantage and demonstrate strong performance on point cloud tasks. The absence of such comparisons makes it difficult to fully demonstrate KAN's unique value relative to “non-MLP efficient architectures.”

2. The paper only experimentally verifies Efficient-KAN's efficiency gains without providing a mathematical explanation for why “rational functions are more suitable for point cloud data than B-splines” or the “basis for selecting the number of channel groups (g)” (Table 1 only presents the parameter formula without justifying the optimal grouping strategy).

3. The paper fails to supplement comparisons of feature responses between Efficient-KAN and native KAN (e.g., local feature responses). or the rationale for selecting the channel grouping number (g)" (Table 1 provides only parameter formulas without demonstrating optimal grouping strategies).

4. Avoid excessive embellishments in your paper presentation. For instance, in the critical Figure 2, all elements are “both bold and italicized”; elements in Figure 4 are partially obscured.

5. To enhance the literature context and strengthen the theoretical-experimental connection in point cloud understanding, it is suggested to supplement the following relevant papers:
[1] Inter-modal masked autoencoder for self-supervised learning on point clouds, IEEE TMM 2023;
[2] Triple Point Masking, IEEE TCSVT 2025.

**Questions:**

1. Benchmark Completeness: Why were recent efficient point cloud architectures like PointMamba and PointRWKV not included in the comparison? Please supplement with a side-by-side comparison of these methods against PointKAN in terms of “Accuracy - Parameters - FPS,” especially highlighting efficiency differences in large-scale point cloud scenarios (e.g., millions of points).

2. Efficient-KAN Design Rationale: Was the choice of channel grouping number (g) validated through systematic experimentation? What is the pattern of how different grouping numbers affect “Accuracy - Parameters - Computational Cost”? Please supplement with ablation experiments on grouping numbers and explain the rationale for the optimal grouping strategy from the perspective of point cloud channel correlation.

3. Fundamental Differences Between KAN and MLP: In point cloud feature extraction, can KAN's “learnable activation functions” demonstrate core advantages over MLP's “fixed activation + stacked linear layers” in capturing geometric features (e.g., feature resolution, gradient propagation efficiency)? Can this be further validated through quantitative metrics (e.g., cosine distance between foreground and background features, gradient norm)?

**Details Of Ethics Concerns:**

No Ethics Concerns

---

### Official Review · Reviewer_3GoS · 2025-10-29

**Soundness:** 3
**Presentation:** 3
**Contribution:** 3
**Rating:** 4
**Confidence:** 4

**Summary:**

This paper addresses the limitations of standard Multi-Layer Perceptrons (MLPs) in point cloud analysis, namely their reliance on fixed activation functions and the high computational cost required to capture complex geometric features. The authors propose PointKAN, a novel hierarchical architecture that replaces MLP blocks with Kolmogorov-Arnold Networks (KANs), which utilize learnable univariate functions (B-splines) for local feature processing. To address the efficiency issues of B-splines, the paper also introduces PointKAN-elite, a variant that uses more efficient rational functions and grouped parameter sharing. Experiments demonstrate its effectiveness.

**Strengths:**

1.	The paper presents a novel and well-motivated application of the recently proposed KAN architecture to the domain of hierarchical 3D point cloud processing. This is a timely research direction that moves beyond traditional fixed-activation MLP structures.
2.	Extensive experimentation and ablation studies validate the effectiveness of the proposed method.
3.	This paper is organized well.

**Weaknesses:**

1.	Though this method has evaluated its performance. It will be more convincing to add comparison with more baselines like PointNeXt[1], Recon[2], ACT[3]，Mamba3D[4]. The paper's central claims of performance and efficiency are undermined by the omission of critical baselines like PointNeXt. PointKAN-elite (93.3% OA, 3.1M params, 2.3G FLOPs)  is less accurate and less efficient than PointNeXt-S (94.0% OA, 1.4M params, 1.6G FLOPs). Furthermore, the proposed models show weak robustness on the challenging ScanObjectNN PB-T50-RS benchmark, where PointKAN-elite (84.1%) underperforms not only PointMLP (85.0%)  but also falls significantly behind PointNeXt (87.7%). This suggests the KAN architecture may be less robust to real-world noise.
2.	It will be interesting to report on the training/inference speed comparison with the baseline model PointMLP-elite.
3.	The related work (Sec 2.1) lacks discussion about many methods, please update.
4.	In Sec 3.2.1, The Group-Norm Mudule is the same as the GEOMETRIC AFFINE MODULE in PointMLP, the authors should apply it in the article instead of claiming it directly.
5.	The justification for the KAN / MLP hybrid design in the main paper (Sec 3.2.3)  is brief and confusing. A much stronger and more intuitive rationale—comparing KAN's local support characteristic (ideal for local geometry) versus MLP's global connectivity (ideal for feature aggregation)—is buried in Appendix D.3. This core methodological justification should be moved from the appendix to the main paper (Sec 3.2) to strengthen the manuscript


[1] Pointnext: Revisiting pointnet++ with improved training and scaling strategies. NeurIPS 2022.

[2] Contrast with Reconstruct: Contrastive 3D Representation Learning Guided by Generative Pretraining. ICML 2023.

[3] AUTOENCODERS AS CROSS-MODAL TEACHERS: CAN PRETRAINED 2D IMAGE TRANSFORMERS HELP 3D REPRESENTATION LEARNING? ICLR 2023.

[4] Han, Xu, et al. "Mamba3d: Enhancing local features for 3d point cloud analysis via state space model." ACMMM. 2024.

**Questions:**

Refer to the Weakness.    I will improve my rates when all the weaknesses are solved.

---

### Official Review · Reviewer_aSiE · 2025-10-30

**Soundness:** 2
**Presentation:** 2
**Contribution:** 2
**Rating:** 4
**Confidence:** 4

**Summary:**

This paper introduces PointKAN, a novel deep learning architecture for 3D point cloud analysis. The core contribution is a hierarchical framework composed of 1) a Geometric Affine Module (GAM) for robust feature normalization, 2) a KAN-based Local Feature Processing (LFP) module to extract local geometric features, and 3) an MLP-based Global Feature Processing (GFP) module for aggregating features. To mitigate the parameter inefficiency of standard KANs, the authors also present PointKAN-elite, a lightweight variant that utilizes rational functions and parameter sharing. The paper claims that PointKAN and its elite version outperform strong MLP-based baselines like PointMLP on standard benchmarks for classification (ModelNet40, ScanObjectNN) and part segmentation (ShapeNetPart), while also offering significant reductions in computational cost.

**Strengths:**

1. The paper applies to the domain of 3D point cloud analysis with a novel hierarchical architecture that thoughtfully combines KANs and MLPs, assigning them to tasks where they are theoretically most suitable (KANs for local feature extraction, MLPs for global aggregation).

2. The introduction of PointKAN-elite addresses the practical concerns of KAN's significant parameter count and computational overhead. The detailed analysis of parameters, FLOPs, and inference speed provides a clear demonstration of the practical advantages of this lightweight version.

3. The authors have conducted experiments across a commendable range of tasks, including shape classification, part segmentation, and few-shot learning.

**Weaknesses:**

1. The paper states that KANs, due to their learnable activation functions, possess a "stronger ability to learn complex geometric features" and exhibit "superior sensitivity to local geometric variations" compared to MLPs. Intuitively, this advantage should be most pronounced in tasks requiring fine-grained local understanding, such as part segmentation. However, the paper's own results contradict this narrative. In Table 4, the proposed PointKAN achieves an instance mIoU of 85.6% on ShapeNetPart, falling behind the 86.1% achieved by the MLP-based PointMLP baseline. The fact that the proposed method underperforms on the very task that should best showcase its theoretical strengths is **a major flaw that undermines the core motivation of the paper**.

2. The title, "KAN or MLP? ..." suggests a definitive **selection** between the two networks. However, the proposed PointKAN architecture is explicitly a hybrid model that strategically uses **BOTH** KAN blocks (for local features) and MLP-based blocks (for global features). This framing of title is misleading and does not accurately reflect the paper's contribution.

3.  The paper contains several technical errors that question its scientific rigor:
    *  In Equation 6, the denominator in the rational function for Efficient-KAN, $Q(x) = \sqrt{1+(b_{1}x+\cdot\cdot\cdot+b_{n}x^{n})^{2}}$, **is not a rational polynomial**. Therefore, the function $\phi(x)$ is also **not a rational** polynomial as claimed.
    *  In Line 296, the authors state that the coefficients $\{a_{i}\}$ have *m* elements, but a polynomial of degree *m* has *m+1* coefficients (from $a_0$ to $a_m$). This error leads to an incorrect parameter calculation for Efficient-KAN in Table 1. The formula provided, $d_{in}\times d_{out}+d_{out}+(n+m\times g)$, appears flawed and does not logically follow from the description of grouped parameter sharing. It may should be revised to $d_{in}\times d_{out}+d_{out}+(n+m+1)\times g$.
    *  In Line 232, the learnable affine parameters $\alpha$ and $\beta$ are defined as being in $\mathbb{R}^{2d}$. However, they are applied via an element-wise Hadamard product to normalized features of dimension $d$. For this operation to be valid, $\alpha$ and $\beta$ must also be in $\mathbb{R}^{d}$.

**Questions:**

Please respond to the comments in Weaknesses.

---

### Official Review · Reviewer_AcBQ · 2025-10-30

**Soundness:** 2
**Presentation:** 3
**Contribution:** 1
**Rating:** 2
**Confidence:** 5

**Summary:**

The main idea of this manuscript is to use KANs instead of MLPs for classification and part segmentation of 3D point clouds. Machine learning experiments on different datasets, such as ModelNet 40 and ScanObject, have been performed, and a comparison between the proposed method and others, such as DGCNN and PointNet-KAN, has been conducted. The proposed method has been inspired by the PointMLP paper (Rethinking Network Design and Local Geometry in Point Cloud: A Simple Residual MLP Framework), of course, now using KANs instead of MLPs.

**Strengths:**

The paper is easy to read and understand. It contains high-quality tables and figures. Mathematical formulations make sense and sound reasonable, without writing so many unnecessary details. At each table, the authors tried to compare different methods from different aspects, which is very informative.

**Weaknesses:**

The main issue and weakness of this paper comes from a lack of a complete literature review. The idea of using KAN instead of MLP in PointNet was presented for the first time one year ago (appeared on arXiv), and later on, the manuscript was published in Computers & Graphics, which you can find the link here:

https://doi.org/10.1016/j.cag.2025.104319

In this manuscript, the authors rarely addressed that paper and also, and they referred to its arXiv version rather than what has been published in Computers & Graphics. Comparing that paper with the current manuscript, we observed that the first one made an experiment on semantic segmentation, which is an absent experiment in this manuscript. Moreover, the author of that paper combined KAN with PointNet++ and also proposed the combined version of KAN and MLP, both embedded in PointNet++. Additionally, in the proposed tables in the manuscript, the authors made a comparison between their own method and old methods, but not with the new methods, which have a better performance than the proposed method. For example, in Table 2, they did not include ShapeLLM, which has a score of 94.8 for ModelNet40. See the paper of ShepeLLM (published 2024):

https://arxiv.org/pdf/2402.17766

So, given that the main idea of this manuscript has already been published in Computers and Graphics, I believe the novelty of the current manuscript is not at the level of ICLR 2026.

As a more general concern, the motivation for using KAN instead of MLP is not clear in the current manuscript, which I think it even shows up in the title of the manuscript: “KAN OR MLP?”. The question mark in the title implies that even the authors do not know why we need to use another one. All in all, while I appreciate the work by the authors, I strongly believe that the manuscript suffers from a lack of novelty, given that the main idea of this work has been published in a reputable journal such as Computers and Graphics.

**Questions:**

Please see the box of "Weaknesses". Thank you.

---

### Note · Authors · 2025-11-13

I have read and agree with the venue's withdrawal policy on behalf of myself and my co-authors.